# Influence of Ovarian Status and Steroid Hormone Concentration on Day of Timed Artificial Insemination (TAI) on the Reproductive Performance of Dairy Cows Inseminated with Sexed Semen

**DOI:** 10.3390/ani13050896

**Published:** 2023-03-01

**Authors:** Stanimir Yotov, Ivan Fasulkov, Anatoli Atanasov, Elena Kistanova, Branimir Sinapov, Boyana Ivanova, Dobri Yarkov, Darina Zaimova

**Affiliations:** 1Department of Obstetrics, Reproduction and Reproductive Disorders, Trakia University, 6000 Stara Zagora, Bulgaria; 2Institute of Biology and Immunology of Reproduction, Bulgarian Academy of Sciences, 1113 Sofia, Bulgaria; 3Department of General Livestock Breeding, Faculty of Veterinary Medicine, Trakia University, 6000 Stara Zagora, Bulgaria; 4Department of Industrial Business and Entrepreneurship, Faculty of Economics, Trakia University, 6000 Stara Zagora, Bulgaria

**Keywords:** cows, ovarian status, steroid hormones, pregnancy, sexed semen

## Abstract

**Simple Summary:**

The use of sexed semen results in a decrease in the number of cows required for progeny testing, the fast production of replacement dairy heifers and a significant reduction in the herd renewal period. Due to some of the disadvantages of sex-sorted semen, such as lower concentration, morphological damage and sperm of lower fertility, there are many problems related to the selection of animals appointed to estrus synchronization programs for artificial insemination with sexed semen. The present study showed a strong relationship between ovarian status, the size of the PF and the estradiol and progesterone concentrations on the day of TAI and the reproductive performance of dairy cows, irrespective of the estrus synchronization treatment used. In this context, the simple and cost-effective ultrasound examination of cows’ ovaries on the day of TAI can be a reliable tool for the selection of candidates for sex-sorted semen application.

**Abstract:**

This study aimed to evaluate the effect of the ovarian status and steroid hormone concentration on the day of TAI on the reproductive performance of dairy cows subjected to estrus synchronization treatment and timed artificial insemination with sexed semen. Seventy-eight cyclic Holstein cows pre-treated with PGF2α-GnRH were divided in two groups—I (Preselect-OvSynch, *n* = 38) and II (OvSynch+PRID-7-day+eCG, *n* = 40)—and inseminated with sexed semen. The presence of preovulatory follicle (PF) with or without corpus luteum (CL), the PF diameter, the estradiol (E_2_) and progesterone (P_4_) concentrations on the day of TAI, the pregnancy rate (PR) and embryo loss were determined. On the day of TAI, 78.4% of all the pregnant cows presented a PF (mean size 1.80 ± 0.12 cm) without CL, low P_4_ (0.59 ± 0.28 ng/mL) and high E_2_ (12.35 ± 2.62 pg/mg) concentrations. The positive correlation between the size of the PF and the level of E_2_ in the pregnant cows from group II was stronger than that of group I (R = 0.82 vs. R = 0.52, *p* < 0.05). The pregnancy rate on day 30 (57.5% vs. 36.8%) and day 60 (50% vs. 26.3%; *p* < 0.05) and the embryo losses (13% vs. 28.5%) showed better effects of treatment in group II. In conclusion, the ovarian status and the steroid hormone concentration on the day of TAI influence the pregnancy rates of dairy cows subjected to estrus synchronization and timed artificial insemination with sexed semen.

## 1. Introduction

Insemination with sexed semen is an intelligent tool to accelerate herd expansion, minimize waste production, improve animal welfare, increase profitability and reduce greenhouse gas emissions from beef and dairy farming [1,2]. Semen sexing is an assisted reproductive technology that allows the sorting of spermatozoa carrying the X- or Y-chromosome and the production of newborn animals of a precisely defined and preferred sex [3,4]. Its main advantage is the production of more female offspring with valuable genetic traits in a shorter period. The use of sexed semen results in a reduction in the number of cows required for progeny testing, the fast production of replacement dairy heifers and a significant reduction in the herd-renewal period. Accelerated genetic progress and the accomplishment of in-house production of replacement heifers increase the degree of farm biosecurity. In line with modern standards of animal welfare, there is a reduction in difficult births when the sexed semen originates from bulls selected for ease of calving [5,6,7,8].

Despite the aforementioned advantages, the concentration of spermatozoa in sexed semen is lower than in conventional semen, and the sorting procedure usually leads to some morphological damage to the sperm cells as well as reduced mitochondrial activity, which is related to low fertility [9,10,11]. The standard dose of one straw with sexed sperm is approximately 2 × 10^6^ sperm, while the straws with conventional semen contain 15–20 × 10^6^ spermatozoa per dose [12,13]. Sperm cell damage is associated with lower survival in the genital organs of female animals (12–16 h for sexed vs. 24–36 h for conventional semen) and requires excellent heat-detection management [14]. Significantly greater conception rates in heifers than in lactating cows and animals with observed hormonally induced estrus have been determined by different authors [15,16,17,18]. Because of this reduced fertility, most of these authors suggest the application of sexed semen for the insemination of heifers with observed estrus. 

Nevertheless, studies about the use of sexed semen in lactating dairy cows are in progress. Maicas et al. [19] report increased pregnancy per artificial insemination (P/AI) in lactating cows when target animals were selected based on base parity, body condition score (BCS), economic breeding index (EBI) and days in milk. In this aspect, DeJarnette et al. [20] recommend artificial insemination with sexed semen mainly in dairy cows with observed estrus, and recommend avoiding the use of a fixed-time insemination, as the success rate was highly variable. In contrast, Karakaya-Bilen et al. [21], comparing three TAI protocols with conventional and sexed semen, conclude that presynchronization increases ovulation and improves the P/AI in cows inseminated with both semen types. According to Darke et al. [22], there is no significant difference in P/AI when cows are inseminated with sexed semen at 16 or 22 h after GnRH treatment, suggesting that 16 to 22 h after GnRH injection probably encompasses the optimal window for the timing of AI with sex-sorted semen. However, Louber et al. [23] underline that cows inseminated with sexed semen 16 h after the Double-Ovsynch program yielded more P/AI than cows inseminated 24 h after the same treatment. The lower result was associated with decreased time for sperm transport and capacitation, as well as differences in the hormonal milieu at critical time points near the end of the synchronization protocol. Guner et al. [24] observed a higher pregnancy rate in cows that expressed estrus following synchronization with progesterone-based OvSynch than cows with no observed estrus. Different studies have reported the influence of estradiol and progesterone concentrations on follicular development, ovulation and the pregnancy rate in cows with natural or synchronized estrus and TAI [25,26,27,28]. 

Nevertheless, questions remain open regarding the effects of animal selection in synchronization programs and TAI with sexed semen, as well as responses to different hormonal treatments and steroid hormone concentrations at the time of AI on the pregnancy rate. 

Inadequate preovulatory concentrations of estradiol in cows around the time of fixed-time insemination was shown to be a reason for lower pregnancy rates associated with improper uterine pH at the time of insemination [29]. A study by Herlihy et al. [26] demonstrated differences in periovulatory E2 concentrations between treatments to synchronize estrus and ovulation. They observed increased periovulatory E_2_ concentrations with a CIDR-based compared with an OvSynch protocol. Preovulatory concentrations of estradiol probably play a crucial role in pregnancy maintenance, not only directly by regulating uterine gene expression but also indirectly throughout the subsequent estrous cycle [27]. 

In most mammals, progesterone production by the CL is essential for the establishment and maintenance of pregnancy and the stimulation of the uterine functions that allow early embryonic development, implantation, placentation and successful fetal and placental development to term [30]. According to Cerri et al. [25] reduced concentrations of progesterone before ovulation can modify endometrial gene expression beyond steroid receptors to a variety of other genes that might be essential to early embryonic survival. They determined that cows with lower plasma concentrations of progesterone and greater concentrations of estradiol had larger ovulatory follicle diameters at the end of the synchronization protocol than cows with high progesterone. Baruselli et al. [31] reported the influence of progesterone concentrations during superovulation protocols on follicular growth, oocyte quality and embryo quality, and the need for adjustments to the protocols depending on the animal category and breed.

The aim of the current study was to determine the effect of the ovarian status and steroid hormone concentration on the day of TAI on the reproductive performance of dairy cows subjected to different estrus synchronization treatments and timed artificial insemination with sexed semen.

## 2. Materials and Methods

### 2.1. Animals

This study was carried out with 78 clinically healthy Holstein cows reared on a commercial dairy farm in Bulgaria, located at latitude of 42.183 N, longitude 25.567 E. The animals were from first to third lactation, weighing 680–700 kg, ≥63 DIM (range 63 to 120 DIM),with an average daily milk yield ≥ 28 kg (range 28 to 34 kg), and located in a loose housing system with individual boxes and a self-locking feed fence. Their feed was with a total mix ration, and their daily diet comprised corn silage, triticale haylage, straw, beer mash, soybean meal, medium-coarse grain soft wheat, corn grain, mineral and vitamin supplement, natrium chloride and a toxin binder with water intake ad libitum. The investigation was conducted between October and December. All procedures were in agreement with the Bulgarian Veterinary Law (25 January 2011) regarding the living conditions and welfare of livestock animals used for experimental purposes, which is adapted to European Union regulation 86/609 (approval no. 3601/09.12.2020).

### 2.2. Experimental Design, Hormonal Treatments, Timed Artificial Insemination, Pregnancy Rate and Embryo-Loss Determination

Animal selection was based on ultrasound examination of reproductive organs and ovarian status determination. A transrectal ultrasonography was conducted using ultrasound scanner SonoScape S2 Vet and a multifrequency (7–12 MHz) linear transducer (SonoScape Medical Corporation. Shenzhen, China). Only cows with corpus luteum (CL) and at least one large follicle (≥10 mm in diameter) in one of the ovaries were included in the study. Their follicular diameters were measured using the in-built scale provided with the ultrasound, and in calculation, only the largest was used. The animals were divided into two groups according to the hormonal treatments, as the proportions of animals that were close in terms of days in milk, average milk yield and number of lactations were almost the same. The estrus synchronization protocols were designed to start on day 6–7 of estrous cycle (Figure 1). After animal selection, both groups were subjected to presynchronization treatment, including single intramuscular injection of PGF2α (25 mg dinoprost tromethamine, Enzaprost T^®^, CEVA Santé Animale, Libourne, France) on the day of the first ultrasound, followed by intramuscular GnRH administration (100 μg gonadorelin as diacetate, Ovarelin^®^, CEVA Santé Animale, Libourne, France) on the day of heat detection (3rd or 4th day). The heat detection after the prostaglandin injection in the presynchronization treatment was registered by an electronic AfiMilk^®^ system for activity monitoring (AfiMilk, Ltd., Kibbutz Afikim, Israel). Day 7 after the end of the hormonal pre-treatment was accepted as day 0 of estrus synchronization protocol. The first group (group I, *n* = 38) was treated with Preselect-OvSynch protocol, including 100 μg GnRH on days 0 and 9 and 25 mg PGF2α on day 7. Timed artificial insemination was carried out 20 h after the last GnRH injection. The second group (group II, *n* = 40) was included in OvSynch+PRID-7-day+eCG protocol. On day 0, a progesterone-releasing intravaginal device (PRID Delta^®^ containing 1.55 g progesterone, CEVA Santé Animale, Libourne, France) was inserted into the vagina and 100 μg GnRH was injected intramuscularly. On day 7, the PRID was removed and each animal received 25 mg PGF2α and 500 UI eCG (equine chorionic gonadotropin; Folligon^®^, Intervet International B.V., Boxmeer, The Netherlands) followed by a second injection of 100 μg GnRH 56 h later. The artificial insemination was 16 h after the second GnRH administration.

The ovarian status (presence of follicles with or without CL in one of the ovaries and diameter of the preovulatory follicle) was determined by ultrasound immediately before TAI (Appendix A). The largest follicle located in the left or the right ovary was defined as a preovulatory follicle (PF). The timed artificial insemination of all cows was performed using frozen sexed semen in straw from the same bull with a high fertility index. The semen was deposited deep cervically by the same operator. The reproductive performance included registration of pregnancy rate and embryo loss. The pregnancy status (pregnant or non-pregnant cows) was determined by transrectal ultrasonography on days 30 and 60 after the TAI. A positive pregnancy diagnosis was recorded upon observation of an echogenic embryo located in echogenic uterine lumen (Appendix A). Embryo loss was calculated according to the pregnancy results on days 30 and 60 after TAI.

### 2.3. Measurement of Steroid Hormone Concentrations

Blood samples were collected on day of TAI before the ultrasound examination. The serum from the coagulated blood samples was separated by centrifugation (3000× *g* for 15 min) and stored in a sterile tube at −20 °C until analysis. Steroid hormone concentrations were measured by an automated quantitative enzyme-linked fluorescent immunoassay (ELFA; VIDAS; ImmunoDiagnostic Assay System, bioMerieux, Craponne, France). For the progesterone (P_4_) assay, we used a VIDAS^®^ Progesterone kit with an analytical sensitivity of 0.1 ng/mL and imprecision within and between runs of 5.7–3.8% and 6.2–3.8%, respectively. The level of estradiol-17β (E_2_) was determined using a VIDAS^®^ Estradiol II kit with an analytical sensitivity of 9 pg/mL and imprecision within and between runs of 7.5–3.9% and 9.5–5.1%, respectively. The measurement range of the kit was 9–3000 pg/mL; therefore, the samples with values < 9 pg/mL were counted as zero.

### 2.4. Statistical Analysis

The results were processed by computer program Statistica version 10.0 (Stat-Soft. Inc. Tulsa, OK, USA). The values of different parameters are presented as mean ± standard deviation (mean ± SD). An analysis of variance for main effects (ANOVA) was used to estimate the statistical significance of the influence of different factors on the likelihood of pregnancy. The strength of the relationship between parameters was assessed by Pearson’s linear correlation test. The significance of the differences between the mean values for different groups and subgroups was compared using the post hoc Tukey’s test or a Chi-squared test for comparison of proportions with small samples. Statistical significance was considered at *p* < 0.05.

## 3. Results

At the beginning of the study, the selected animals had corpus luteum, small or medium follicles and at least one large follicle located in one of the ovaries, which was accepted as an indicator of cyclicity. All cows had a positive response after the PGF2α injection of the presychronization treatment, evidenced by the detection of estrus behavior between 3 and 4 days later.

The comparative analysis of the ovarian status showed that in group I, 16.6% more non-pregnant cows had PF with CL on the day of TAI, while of the pregnant animals, 28.6% more had PF without CL. (Table 1). In group II, a significantly higher percentage of PFs without CL on the day of TAI was observed in both the non-pregnant and the pregnant animals (70.6% and 87%; *p* < 0.05).

The mean diameters of the PFs in the non-pregnant and pregnant cows treated with Preselect-Ovsynch were close (*p* = 0.12). The value of this parameter in the OvSynch+PRID+eCG group was higher in the pregnant than in the non-pregnant animals (1.88 ± 0.10 mm vs. 1.68 ± 0.19 mm; *p* < 0.05). The mean PF diameter in the pregnant cows in group II also differed significantly (*p* < 0.05) from the follicular diameters in the pregnant and non-pregnant cows in group I. 

The blood progesterone concentrations on the day of TAI indicated significant (*p* < 0.05) differences between the pregnant and non-pregnant dairy cows, irrespective of the estrus synchronization protocol used. This was clearly expressed in group I, where the concentration of P_4_ in the non-pregnant animals exceeded that in the pregnant animals by a few times (Table 1). The lowest level of P_4_ on the day of TAI was recorded in the pregnant cows from group II. The mean estradiol-17β concentrations in the pregnant and non-pregnant cows in both groups also differed significantly (*p* < 0.05). The highest level of E_2_ on the day of TAI was determined in the pregnant animals that received OvSynch+PRID+eCG treatment. It was higher compared to the measured level E_2_ in the non-pregnant animals in this group (*p* < 0.05) and the E_2_ values in the whole group I (Table 1). 

The pregnancy rate on day 30 tended to be higher by about 20% in the OvSynch+PRID+eCG- compared to the Preselect-OvSynch-treated cows, but the difference was not significant (*p* = 0.16). However, on day 60, the PR values were significantly greater in group II compared to group I (Table 1). In addition, the embryo losses were 15.5% lower in the animals treated with protocol II than in those treated with protocol I. The animals with embryo loss in pregnant subgroup I had PF only, and PF and CL on the day of TAI at rates of 25% and 75%, respectively. At the same time, 100% of the cows with embryo loss in the pregnant subgroup II exhibited PF and CL. From a total of seven animals, PF with a Cl was detected in 85.7% of cows, while in 14.3%, this was not observed. The average diameter of the PFs on the day of TAI was 1.76 ± 0.07 mm. The pregnancy reductions of 10.3% in group I and of 7.5% in group II between 30 and 60 days were not statistically significant. 

The investigated effects, irrespective of the estrus synchronization treatment used, are presented in Table 2. Comparative analysis showed that overall, the pregnant cows had significantly more cases of PF without CL in the ovaries on the day of TAI. The mean values for the PF diameter and E_2_ concentration on the day of TAI were higher in the pregnant group than in the non-pregnant group (*p* < 0.05), whereas the concentration of P_4_ was lower (Table 2. *p* < 0.05). A strong correlation was observed between the sizes of the preovulatory follicles and the level of E_2_ in all the animals (R = 0.93, *p* < 0.001), as well as in the pregnant animals treated with both synchronization schemes (R = 0.79, *p* = 0.001). Interestingly, in the pregnant cows from group II, this correlation was stronger compared to that in the pregnant cows from group I (R = 0.82 vs. R = 0.52, *p* < 0.05). The analysis (Appendix A) showed that both the PF size and the E_2_ concentration were influenced by the ovarian status (*p* = 0.007) and by the treatment protocol (*p* = 0.019). The observed power of the effect of pregnancy status on these parameters was the highest (observed power 0.94, *p* < 0.05).

## 4. Discussion

The success of the artificial insemination of dairy cows with sexed semen is rather variable and depends on different on-farm factors, the age, BCS and parity of the animals, sire selection, service number, estrus detection management and cow fertility potential [32,33,34]. A reduced number of spermatozoa per dose, the limited lifespan of sorted sperm cells, the insemination of cows with inadequate development of ovulatory follicles, or failed ovulation due to a hormonal imbalance and the performance of artificial insemination at an inappropriate time are the main reasons for low conception rates [35,36,37,38]. In this context, some authors recommend eliminating some of these problems by including dairy cows in estrus synchronization programs with fixed times of artificial insemination [39,40,41]. A major advantage of these programs is the possibility of timed AI, which is less time-consuming, makes work with the animals easier and provides control over ovulation time [42]. Starting the OvSynch treatment between days 5 and 9 of the estrus cycle is associated with a greater probability of having ovulations with a new CL and the development of a new follicular wave. This results in the synchronized ovulation of a newly formed dominant follicle at the end of the synchronization program and better conception rates [43,44]. Different studies reveal a positive effect of progesterone-releasing devices on conception rates after their inclusion in estrus synchronization protocols [21,24,45].

This study presents results regarding the effects of ovarian status and steroid hormone concentrations at the time of TAI on the pregnancy rates of dairy cows treated by Preselect-OvSynch and OvSynch+PRID+eCG estrus synchronization protocols, starting on day 6–7 of the estrous cycle and timed AI with sexed semen from a high-fertility bull. The uniform distribution of the animals in both treatment groups was proven by the fact that the same ovarian status was observed before the start of the experiment, and by the positive response of all the cows to the presynchronizing PGF2α-GnRH treatment which was evidenced by heat detection 3 to 4 days later. The strategy for presynchronization, involving the induction of ovulation 6–7 days before the start of the estrus synchronization protocol, results in more animals having ovulations after the first GnRH injection of the protocol [46]. If cows fail to ovulate after the first GnRH injection, the lifespan of the preovulatory follicle at the end of the synchronization protocol is prolonged, which is associated with the ovulation of aged oocytes, reduced embryo quality and low pregnancy rates [47,48]. 

Nevertheless, we considered the ovarian status and the concentration of steroid hormones on the day of TAI as determining factors in the successful ovulation and the achievement of an acceptable PR in the use of sexed semen. Most of the non-pregnant animals in group I (58.3%) had PF and CL on the day of TAI, while the majority of the animals that became pregnant (64.4%) bore PF without CL. This information and the significant (*p* < 0.05) difference in the ovarian status on the day of TAI between the different subgroups of cows in group II supported the aforementioned hypothesis. The absence of CL on the day of TAI in more than 70% of the pregnant animals after the OvSynch+PRID+eCG treatment was in agreement with other reports of the better synchronizing effect of progesterone-based protocols in comparison with conventional OvSynch. 

The phenomenon behind the reduction in the efficiency of conventional OvSynch protocols may be incomplete luteolysis after the PGF2α injection, followed by inadequate levels of progesterone and the suppression of the second follicular wave, which makes the synchronization of ovulation and TAI impossible [23,49]. This is crucial for successful insemination with sexed semen because of the limited lifespan of the spermatozoa in the female genital organs, the reduced number of sperm per straw, and the potential pre-capacitation induced by the sorting procedure [14,50]. Different researchers stated that progesterone supplementation before timed AI imitates P_4_ concentration in the late luteal phase of a regular estrous cycle, leading to an increase an ovulation synchrony and P/AI [51,52,53]. This information is in agreement with the better pregnancy rate in group II observed in the current experiment.

Our hypothesis supposes that the diameters of the preovulatory follicles and steroid hormone concentrations can affect ovulation and, consequently, pregnancy after AI with sexed semen. The large size of PFs (in the range of 1.72 cm to 1.88 cm) was associated with a positive effect on the ovulation and PR in both groups. The determination of the highest mean diameter of the PFs in the pregnant animals from the second group and the relative enhancement of the sizes of the preovulatory follicles in the non-pregnant cows from the first group may be explained by the administration of eCG during the PGF2α injection. The influence of follicular size near the time of TAI on pregnancy success is still debatable. Bello et al. [54] determined the best conception rates after the ovulation of a 1.6 cm follicle, whereas conception rates decreased if the preovulatory follicle was either smaller or larger than 1.6 cm. According to Colazo et al. [55], after the second GnRH injection in synchronization programs with or without P4 supplementation, there was a variable range of ovulatory follicle sizes between lactating dairy and beef cows. They reported that follicles between 1.7 cm and 2.5 cm in diameter retain their capacity to ovulate, and cows with ovulatory follicles greater than 2.0 cm have an increased probability of pregnancy loss. Conversely, Berg et al. [56] found that follicles smaller than 1.5 cm ovulated significantly later, while cows with medium-sized and large follicles ovulated earlier than other animals. These controversial results presume that follicular size is not the only factor responsible for successful ovulation and conception. 

In the current study, estrus synchronization treatments had a considerable effect on the functional status of the ovarian structures on the day of TAI. They were associated with the production of progesterone and estrogen from the CL and the preovulatory follicle, respectively. The significant (*p* < 0.05) difference in the mean P_4_ concentration between the non-pregnant and pregnant animals subjected to both treatments was probably due to the high percentage of animals bearing functional and active CL in the non-pregnant group at the time of TAI, caused by failed or incomplete luteal regression. The lower concentration of P_4_ determined in the pregnant animals from group II can be attributed to better luteolysis after the application of the Ovsynch+PRID+eCG protocol. In this regard, Garcia-Muñoz et al. [57] observed insignificant differences between treated and control groups after the administration of exogenous progesterone around the time of ovulation, and treatment with d-cloprostenol 120 h after ovulation. It is still not clear whether exogenous progesterone influences the sensitivity of the cow CL to PGF treatment. Although the same number of animals presented CL at the time of TAI, we observed a slight increase in P_4_ in non-pregnant subgroup II compared to pregnant subgroup I. One reason for this result could be the residual quantities or the delayed clearance of the supplemented progesterone. Cerri et al. [58] also reported augmented levels of P_4_ observed at the time of TAI after PRID removal and the first PGF_2_ alfa administration. 

The significant difference (*p* < 0.05) in the mean E_2_ concentration between the pregnant and non-pregnant animals of the treated groups was an indicator of the increased steroid activity of the larger preovulatory follicles, which was confirmed by correlative analysis. This effect was clearly visible in group II, where the highest E_2_ concentration corresponded with the largest size of the preovulatory follicles. A similar increase in E_2_ in the larger PFs was observed in group I. On the other hand, the preovulatory follicles formed after Preselect-Ovsynch or Ovsynch+PRID+eCG treatment probably had different levels of estradiol production. Despite the insignificant difference between the mean diameters of the PFs in both subgroups subjected to Preselct-Ovsynch, a significantly (*p* < 0.05) higher mean concentration of estradiol-17β in the pregnant subgroup was observed. In this respect, Bridges and Fortune [59] emphasized that extended progesterone exposition after the incomplete regression of the CL had a negative indirect effect on estradiol-17β production. Herlihy et al. [26] also reported a positive correlation between increased preovulatory follicle size and circulating concentrations of E_2_ in progesterone-based synchronization protocols due to a longer period of preovulatory follicle growth. After P_4_ withdrawal, an increase in LH pulse frequency and mean LH concentrations promoted follicle growth and increased estrogen production. The second GnRH caused a rapid decrease in circulating estrogen concentrations, and OvSynch had less peak preovulatory E_2_ than the CIDR-TAI protocol. An additional factor in the increase in steroid activity in the preovulatory follicles of group II was probably the PMSG stimulation. According to Hirako et al. [60], the administration of 500 UI eCG in cattle lead to a significant increase in the release of estradiol-17β.

Despite the lack of significant differences in PR on day 30, the PR tended to increase after the Ovsynch+PRID+eCG treatment compared to the Preselect-Ovsynch treatment (38.6% vs. 57.5%; *p* = 0.09). The lower percentage of pregnant cows in group I, which was probably due to incomplete luteolysis, corresponded with the high P_4_ level on the day of TAI, which results in reduced pulsatile LH release and negatively affects final follicle maturation, oocyte maturity, and gamete transport [59,61]. Martins et al. [62] determined that complete functional luteolysis had a strong positive relationship with the time of ovulation. The lack of complete regression of the corpus luteum resulted in a low P/AI because of the increased progesterone concentration close to the time of AI in cows subjected to conventional Ovsynch treatment [63]. The higher pregnancy rate in group II can be explained by the inclusion of exogenous progesterone and eCG in the second protocol. In line with the current study, Colazo et al. [52] determined that the inclusion of PRID in the conventional Ovsynch protocol had a positive effect on P/AI. In fact, progesterone-releasing devices (CIDR or PRID) provide a slow release of progesterone, which regulates the number of estrogen receptors in the medial basal hypothalamus, increasing the responsiveness to estradiol, leading to a preovulatory LH surge. The withdrawal of the progesterone device leads to decreased progesterone and increased LH pulse frequency and mean concentrations of LH, which are associated with an increased number of LH receptors in the granulosa and theca cells, estradiol production by the follicle, and the stimulation of LH surges, thus resulting in ovulation [47,64,65]. Additional evidence for the positive influence of the Ovsynch+PRID+eCG treatment on pregnancy was the registration of a higher percentage of PR in the second group on day 60 (*p* < 0.05).

The pregnancy rate obtained with the Preselect-Ovsynch on day 30 after TAI (36.8%) was close to the 35.5% recorded by Karakaya-Bilen et al. [21], but lower than the 49.0% obtained by Drake et al. [22]. Lauber et al. [23] recorded 50% P/AI in presynchronized dairy cows on day 34 after double Ovsynch treatment and TAI with sexed semen. At the same time, the P/AI in Ovsynch+PRID+eCG (57.5%) was greater than the values observed in primiparous (49.5%) and multiparous (47.6%) lactating cows treated with a PRID-based protocol and inseminated with sexed semen [24]. In agreement with this study was Brozos et al. Ref. [60]’s finding of 54.3% P/AI after the use of PGF2α-GnRH pre-treatment and PRID-based protocols during the winter season. Furthermore, the sexed semen for the AI in our study originated from a bull selected for its high fertility. Drake et al. [22] investigated the interaction between treatment with sex-sorted semen and P/AI according to the bull fertility and detected deviations in the P/AI from 38.2% to 60.3%.

Similar to the findings of previous authors, we also observed decreasing PR between both consecutive ultrasound examinations for pregnancy, which was an indicator of late embryo loss. The embryo loss after the use of both synchronization protocols (28.5% and 13%) corresponded with the data obtained by Karakaya-Bilen et al. [21] and Guner et al. [24]. In disagreement with the obtained data was the embryo loss (5% and 6%) in the dairy cows on days 34 ± 3 and 80 ± 17 determined by Lauber et al. [23]; however, that study used a different estrus synchronization treatment. The higher percentage of embryo loss in group I can be attributed to the ovulation of smaller follicles at the time of TAI and the formation of CL, releasing less P_4_, which is critical for embryo development through the early gestation period [66,67]. In this respect, the diameter of the PFs on the day of TAI in the animals with pregnancy loss in subgroup I (≤1.7 mm) was below the calculated average size (1.8 ± 0.12 mm) of the PFs for all the pregnant animals. 

The assertion of the effect of the ovarian status was supported by the significantly (*p* < 0.05) higher percentage of animals that became pregnant when PF without CL and diameters between 1.68 cm and 1.88 cm on the day of TAI were presented. It is known that the large luteal cells of the future CL, producing 80 to 90% of the progesterone, originate from granulosa cells [68]. Larger ovulatory follicles provided a greater number of granulosa cells, which resulted in a larger CL volume on day 7, producing enough progesterone for adequate pregnancy maintenance [66]. 

The difference (*p* < 0.05) between the mean values of the steroid hormones on the day of TAI between the non-pregnant and pregnant cows was an indicator of the significant effect of the E_2_ and P_4_ concentrations on the pregnancy results. A low P_4_ concentration at TAI in pregnant animals was also recorded by Souza et al. [69]. They observed that a blood progesterone level > 0.5 mg/mL at a time close to insemination decreases fertility by about 50%, while the P_4_ dropping to <0.5 ng/mL 2 days after the GnRH injection suggested a greater probability of pregnancy. Denicol et al. [70] underline the importance of the endocrine environment during follicular growth and the maturation of viable oocytes, and accept the size of the ovulatory follicle as a vital component in the probability of pregnancy. An inadequate endocrine profile at the time of TAI was shown to decrease oocyte quality, gamete transport, fertilization and the uterus’s ability to maintain pregnancy [71,72]. According to Brozos et al. [60], even after the second PGF_2_ treatment in the PRID-based protocol, a considerable number of cows had high progesterone (>1 ng/mL) at the time of AI; this occurred without a decrease in P_4_ concentration in the period between the first PGF_2_ injection and the AI. 

All the aforementioned studies assumed that the ratio between estrogen and progesterone on the day of TAI is a very important factor in successful conception in synchronized dairy cows inseminated with sexed semen. Future detailed investigations with a large number of cyclic cows can clarify this matter.

## 5. Conclusions

The ovarian status and steroid hormone level on the day of TAI significantly affected the reproductive performance of dairy cows subjected to estrus synchronization and artificial insemination with sexed semen. The presence of a large PF, the production of more estrogen, a lack of CL, and low progesterone levels at the time of AI are factors that ensure acceptable pregnancy rates. The application of presynchronization treatment with the Ovsynch+PRID+eCG protocol can additionally improve pregnancy results after AI with sex-sorted semen. The confirmed strong correlation between the values of the PF and E_2_ concentrations allows ultrasound examination of the ovarian status of cows on the day of TAI to be recommended as a reliable tool for the selection of candidates for insemination with sexed semen. This will improve the utilization of sex-sorted semen from bulls with high genetic value, and accelerate genetic progress. 

## Figures and Tables

**Figure 1 animals-13-00896-f001:**
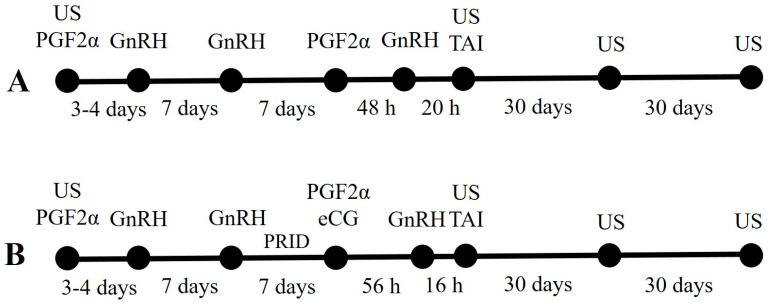
Preselect-OvSynch (**A**) and OvSynch+PRID+eCG (**B**) estrus synchronization protocols in dairy cows (US—ultrasound examination; PGF2α—prostaglandin F2 alfa; GnRH—gonadotropin-releasing hormone; PRID—progesterone-releasing intravaginal device; TAI—timed artificial insemination).

**Table 1 animals-13-00896-t001:** Ovarian status and steroid hormone concentration on day of TAI, pregnancy rate (PR) and embryo loss according to different estrus synchronization treatments.

Parameters	Group I * (*n* = 38)		Group II ** (*n* = 40)	
	Non-Pregnant Subgroup (*n* = 24)	Pregnant Subgroup (*n* = 14)	Non-Pregnant Subgroup (*n* = 17)	Pregnant Subgroup (*n* = 23)
Ovarian status				
Preovulatory follicle with corpus luteum %, (n/n)	58.3 (14/24)	35.7 (5/14)	29.4 (5/17) ^1^	13 (3/23) ^1^
Preovulatory follicle without corpus luteum %, (n/n)	41.7 (10/24)	64.3 (9/14)	70.6 (12/17) ^2^	87 (20/23) ^2^
Diameter of preovulatory follicle (mean ± SD, cm)	1.60 ± 0.24 ^a^	1.72 ± 0.20 ^a^	1.68 ± 0.19 ^a^	1.88 ± 0.10 ^b^
Steroid hormone concentrations				
Progesterone (mean ± SD, ng/mL)	2.35 ± 3.07 ^a^	0.68 ± 0.26 ^b^	0.82 ± 0.24 ^b^	0.52 ± 0.12 ^c^
Estradiol-17β (mean ± SD, pg/mL)	7.32 ± 0.80 ^a^	10.24 ± 0.36 ^b^	10.04 ± 2.12 ^b^	12.86 ± 2.26 ^c^
PR on day 30 %, (n/n)	36.8 (14/38)		57.5 (23/40)	
PR on day 60 %, (n/n)	26.3 (10/38) ^a^		50 (20/40) ^b^	
Pregnancy loss %	28.5 (4/14)		13 (3/23)	

* Preselect-OvSynch protocol, ** OvSynch+PRID-7-day+eCG protocol. Values with different superscript letters within a row are different at *p* < 0.05. Values with different superscript numbers within a column are different at *p* < 0.05.

**Table 2 animals-13-00896-t002:** Ovarian status and steroid hormone concentration on day of TAI according to pregnancy status of the cows.

Groups	Non-Pregnant (*n* = 41)	Pregnant (*n* = 37)
Ovarian status on day of TAI		
Preovulatory follicle with corpus luteum %, (n/n)	46.3 (19/41) ^a^	21.6 (8/37) ^b,1^
Preovulatory follicle without corpus luteum %, (n/n)	53.7 (22/41) ^a^	78.4 (29/37) ^b,2^
Diameter of preovulatory follicle (mean ± SD, cm)	1.64 ± 0.26 ^a^	1.80 ± 0.12 ^b^
Steroid hormone concentrations		
Progesterone (mean ± SD, ng/mL)	1.58 ± 1.06 ^a^	0.59 ± 0.28 ^b^
Estradiol-17β (mean ± SD, pg/mL)	8.68 ± 1.82 ^a^	12.35 ± 2.62 ^b^

Values with different superscript letters within a row are different at *p* < 0.05. Values with different superscript numbers within a column are different at *p* < 0.05.

## Data Availability

All data are included in the present article and its Appendix A.

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
