# Peer review of "Influence of Ovarian Status and Steroid Hormone Concentration on Day of Timed Artificial Insemination (TAI) on the Reproductive Performance of Dairy Cows Inseminated with Sexed Semen"

_animals, 2023, doi:10.3390/ani13050896_

Round 1

Reviewer 1 Report

The authors address an interesting scientific topic sought by farmers regarding the conditions for using sexed sperm for genetic progress and increased profitability.

The manuscript complies with the technical drafting conditions and falls within the specifics of the journal and the special volume.

L. 15: I don't understand this expression. Like AI with sexed, it shortens the interval from calving to lactation. Bring some clarified details, please.

In the introduction, the authors describe and deepen the subject of AI biotechnology very well, citing 22 justified references, but some information is also needed about the role and effects of E and P4 steroids and their relationship with the conception rate.

L. 121: how were the cows selected according to this criterion? As a prediction - being in the luteal phase with the existence of a CL? or was it followed the cows in heat were not AI and were kept for this study?

Regarding material and method and fig 1. :

I think the more appropriate term would be Preselect-OvSynch, use the newer term eCG instead of PMSG. Clarify the moment of PG application on day 3 or 4 - you must be more precise.

L126: All the cows (n=78) taken in the study were in heat after 7 days, from pre-synchronization? - specify.

L130, 137:  About AI specify if: AI was performed by the same operator? to both batches? by which classic TC method or other new devices?

At the results, arguments and explanations should be given as to why in batch 1 the conception rate was lower than in L2. However, the P4 level is above the limit of 1 mg and the FD diameter did not exceed 1.6 cm.

L265: On the day of the first detection of pregnancy - day 30, did CL persist on the ovaries in non-pregnant cows? can you specify

L308: Following the results, can you suggest that after an OvSynch protocol a double dose of PG be administered? for the correction of the CL involution, the decrease of P4, the increase of the PF of E2 and obtaining a qualitative oocyte...

L 417: change the word cows to Cyclic cows

At the S1B ultrasound - I don't think it's a functional CL

Attached you will find the article with other suggested changes.

After correction it can be published.

Author Response

Dear reviewers,

 First, we highly appreciate your positive evaluation of the scientific content of our work.  Also, we thank you for the useful comments. They gave us some positive criticism for reviewing and improving our manuscript.

Considering all your notices, we have made several changes that are included in the revised manuscript and marked in yellow.

Response to Reviewer 1

  1. L. 15: I don't understand this expression. Like AI with sexed, it shortens the interval from calving to lactation. Bring some clarified details, please.

We apologize about inconvenience of the sentence. We accept this remark and the expression was excluded from the text.

  1. In the introduction, the authors describe and deepen the subject of AI biotechnology very well, citing 22 justified references, but some information is also needed about the role and effects of E and P4 steroids and their relationship with the conception rate.

Your remark is valuable. We added to the Introduction some citations discussed the relationship between of E2 and P4 steroids and conception rate.

  1. L. 121: how were the cows selected according to this criterion? As a prediction - being in the luteal phase with the existence of a CL? or was it followed the cows in heat were not AI and were kept for this study?

Animal selection was made by the next manner: All cows were with at least one exhibited estrus before the pre-treatment but did not inseminated. On day of the first ultrasound examination (the start of the hormonal pre-treatment) only cows with corpus luteum and at least one large follicles (≥ 10 mm in diameter) in one of the ovaries were used and treated with PGF2alfa. After that the animals were monitored to estrus expression by electronic Afimilk system and in case of estrus detection they were treated with GnRH for induction of ovulation. This day was accepted as day 0 and the estrus synchronization protocol started seven days later.

  1. Regarding material and method and fig 1: I think the more appropriate term would be Preselect-OvSynch, use the newer term eCG instead of PMSG. Clarify the moment of PG application on day 3 or 4 - you must be more precise.

We accept the suggestion for change of the abovementioned terms. Probably you have in mind the first GnRH ingection. PG injection was made on day of ultrasound examination. Some of the selected animlas were detected in estrus on Day 3 and other on Day 4 after the first PG injection. GnRH was administrated on the day of estrus detection (Day 3 or 4). Because of that in the figure 1 is showed 3-4 day.

  1. L126: All the cows (n=78) taken in the study were in heat after 7 days, from pre-synchronization? - specify.

The animals were not in heat 7 days after the presynchronization treatment. The design was they to be on day 7 of estrus cycle. Probably the sentence The heat detection after the presynchronization treatment was registered by electronic AfiMilk® system for activity-monitoring (AfiMilk, Ltd, Israel)“ is not so clear and we made a change as follow: „The heat detection after the prostaglandin injection in the presynchronization treatment was registered by electronic AfiMilk® system for activity monitoring (AfiMilk, Ltd, Israel)“.

  1. L130, 137:  About AI specify if: AI was performed by the same operator? to both batches? by which classic TC method or other new devices?

From a practical point of view AI of so much animals at same day is very difficult to be done by the same operator. Moreover, it will spend a lot of time and some cows will be out of the appropriate time for insemination. Because of that, the treatment in both synchronized groups did not started at the same time. The protocols were designed in a manner allowing the insemination to be in different days. In addition, not all cows from the same group were inseminated at the same day because some of them exhibited estrus on Day 3 and other on Day 4 after the first PG injection of the pre-treatment, that allowed AI in the same group to be performed in 2 days. We did not use new devices as AlfaVision or Eye breed because of a loss of time when more animals have to be inseminated. We used classical TC method and the sexed semen was deposited by the same experienced operator, but in different days.

  1. At the results, arguments and explanations should be given as to why in batch 1 the conception rate was lower than in L2. However, the P4 level is above the limit of 1 mg and the FD diameter did not exceed 1.6 cm.

The explanation of the lower conception rate in group I was the combination from lower size of the PFs with decreased estrogen production and increased P4 level at the time of AI, compared to group II. The follicles with lower size produced lesser estrogens. The low E2 levels induce inadequate GnRH and LH surge, associated with lack of delay of ovulation and low conception rate, respectively. Moreover, the larger follicles ensure better function of the CL after ovulation, because more granulosa cells will give more luteal cells in the future CL. In addition, the incomplete regression of CL leading to high P4 level at the time of AI can supress GnRH and LH surge, resulting in a lower pregnancy rates. All aforementioned was a reason the ratio between estrogens and progesterone on day of TAI to be showed as an important factor for succesfull conception after AI with sexed semen.

  1. L265: On the day of the first detection of pregnancy - day 30, did CL persist on the ovaries in non-pregnant cows? can you specify.

The information of L265 “Most of the non-pregnant animals in group I (58.3%), had PF and CL on day of TAI, while greater part of the animals became pregnant (64.3 %), beared PF without CLrefers to the ovarian status on Day of TAI. Unfortunately, information about persisting of CL on day 30 is not available. Each animal exhibited estrus behavior after TAI with sexed semen was firstly examined for pregnancy and in lack of pregnancy was re-inseminated with conventional semen. On days 30 and 60 only check for pregnancy was made. The ovarian status of non-pregnant animals was not recorded. 

  1. L308: Following the results, can you suggest that after an OvSynch protocol a double dose of PG be administered? for the correction of the CL involution, the decrease of P4, the increase of the PF of E2 and obtaining a qualitative oocyte...

Thank you very much for your suggestion. It will be presented. Definitely a double dose of PG 12 or 24 hours apart will have a positive effect.

  1. L 417: change the word cows to Cyclic cows

The change was made.

  1. At the S1B ultrasound - I don't think it's a functional CL

The presented luteal structure is regressing CL detected in non-pregnant cow defined with incomplete luteolysis on day of TAI on the base of P4 level. The name of CL was corrected in the figure legend.    

  1. I notice a difference in the size of the text

We checked all text and removed the different size of text.

Reviewer 2 Report

Animals-2175495

Ovarian status and steroid hormones concentration on day of timed artificial insemination (TAI) influence on the reproductive performance of dairy cows inseminated with sexed semen

S. Yotov, et al.

Summary:  The investigators evaluate P4 and E2 concentrations, follicle size at ovulation/CL presence, pregnancy rates at days 30 and 60 and pregnancy loss between day 30 and 60 when using 2 TAI synchronization schemes in hairy cows.  Pregnancy rates were greater in the group receiving a P4 PRID than those receiving Ov Sync protocol.  Regardless of synchronization treatment approximately 80% of animals pregnant at day 30 after TAI exhibited larger preovulatory follicles and higher E2 values, no CL and low P4 levels compared to the non-pregnant group. 

Critique: This was a difficult paper to read.  There are numerous language and grammatical issues throughout the manuscript that need to be corrected.   I would recommend a complete rewrite for correct English grammar prior to a resubmission.

The topic is interesting and timely as many are starting to observe issues with pregnancy loss when using TAI and the percentages of loss observed in this study tend to strongly agree with the limited data on the topic.  The experimental design is sound; however, larger animal numbers would help in this (or future) studies.  The main issue with the data presented in the E2 values.  The sensitivity of the assay is 9 pg/mL with inter- and intra-assay coefficients of 7.5-3.9 (not sure what this means) and 9.5-5.1% (not sure what this means).  Therefore the E2 values presented in table 1 are at or below E2 assay sensitivity.  This should be addressed and one suggestion is to emphasize the preovulatory follicle data. 

I think the authors should also address the pregnancy loss data more thoroughly in table 1 and in the discussion.  For instance, were the animals that lost pregnancies in the subgroups which exhibited a preovulatory follicle and a CL, exhibited a preovulatory follicle only, or both.   In the discussion, the authors state that the amount of pregnancy loss was due to smaller ovulatory follicles and subfunction CL after TAI.  They should include the data they have which should support this statement. 

Other items

Line 15 and 49: How does using TAI and synchronization in general, shorten the time from calving to first lactation?  Should this be shorten the interval to calving and first lactation?

Line 74: and

Line 294:  change to synchronization programs with or without P4 supplementation.

Line 306: due

Line 308: concentration

Line 375: findings

Recommendations:  Reject and request re-submission

Author Response

Dear reviewers,

 First, we highly appreciate your positive evaluation of the scientific content of our work.  Also, we thank you for the useful comments. They gave us some positive criticism for reviewing and improving our manuscript.

Considering all your notices, we have made several changes that are included in the revised manuscript and marked in yellow.

Response to Reviewer 2

  1. Critique: This was a difficult paper to read.  There are numerous language and grammatical issues throughout the manuscript that need to be corrected.   I would recommend a complete rewrite for correct English grammar prior to a resubmission.

Taking into account the suggestion for English improvement, we used English editing service at MDPI.

  1. The topic is interesting and timely as many are starting to observe issues with pregnancy loss when using TAI and the percentages of loss observed in this study tend to strongly agree with the limited data on the topic.  The experimental design is sound; however, larger animal numbers would help in this (or future) studies.  The main issue with the data presented in the E2 values.  The sensitivity of the assay is 9 pg/mL with inter- and intra-assay coefficients of 7.5-3.9 (not sure what this means) and 9.5-5.1% (not sure what this means).  Therefore the E2 values presented in table 1 are at or below E2 assay sensitivity.  This should be addressed and one suggestion is to emphasize the preovulatory follicle data. 

The showed coefficients 7.5-3.9% and 9.5-5.1% indicate the percentages of the deviation when several samples were tested in the same run (within) and when the samples were tested singly in different runs (between). Really, the analytical sensitivity of used kit is 9 pg/mL (measurement range 9-3000 pg/mL), so the samples with lower than 9 pg/mL level were counted as a zero. However, the statistical means were calculated for whole number of animals in groups, which lead to obtain the meaning under the level of analytical sensitivity – 9 pg/mL, in particular, in non-pregnant group. This information is added to material and methods.

  1. I think the authors should also address the pregnancy loss data more thoroughly in table 1 and in the discussion.  For instance, were the animals that lost pregnancies in the subgroups which exhibited a preovulatory follicle and a CL, exhibited a preovulatory follicle only, or both.   In the discussion, the authors state that the amount of pregnancy loss was due to smaller ovulatory follicles and subfunction CL after TAI.  They should include the data they have which should support this statement.

Thank for your valuable remark. The information and explanation of the data about pregnancy loss was added in text.

Other items

  1. Line 15 and 49: How does using TAI and synchronization in general, shorten the time from calving to first lactation?  Should this be shorten the interval to calving and first lactation?

We accept this remark and the sentence was excluded from the text.

  1. Line 74: and

The correction was made. 

  1. Line 294:  change to synchronization programs with or without P4 supplementation.

 The change was made.

  1. Line 306: due; Line 308: concentration; Line 375: findings

The printing errors on the mentioned above lines were removed.

Reviewer 3 Report

Dear autors,

The idea of the work is quite interesting, but the objectives were broader than the number of animals to allow strong conclusions.

This is the main point of not indicating the article for publication.

Author Response

Dear reviewers,

 First, we highly appreciate your positive evaluation of the scientific content of our work.  Also, we thank you for the useful comments. They gave us some positive criticism for reviewing and improving our manuscript.

Response to Reviewer 3

Dear autors,

The idea of the work is quite interesting, but the objectives were broader than the number of animals to allow strong conclusions.

This is the main point of not indicating the article for publication.

  1. We agree with critical remark related to small number of subjects. However, the existing statistical methods for small populations allow estimating the significance of the obtained results. In addition, in accordance with animal ethic issues and farm specific we try to keep not big, but an optimal number of animals that allows making strong conclusion on the base of robust statistic.

Round 2

Reviewer 2 Report

The only suggestion I have for this revised version is beginning on line 235, inclusion of the actual numbers with the percentages of pregnancies loss may be helpful to the reader.

Reviewer 3 Report

Dear autors,

In the first evaluation, it was reported that the small number of animals was the main point of not indicating the article for publication. However, unfortunately, it is still not the only problem. The small number of animals per group was further subdivided (pregnant and non-pregnant sub group). 

The protocols used are totally different. In protocol 2, PMSG was inserted, the second GnRH was applied 8 hours later and TAI was performed 4 hours earlier compared to protocol 1. It is confused.

In view of this, unfortunately, there is no way to recommend the publication of the article in this journal.